# communications
# engineering

# Towards automated sleep-stage classification for adaptive deep brain stimulation targeting sleep in patients with Parkinson's disease

Katrina Carver[1], Karin Saltoun [2], Elijah Christensen[3], Aviva Abosch[4,9], Joel Zylberberg [5,9] & John A. Thompson [6,7,8,9✉]

Sleep dysfunction affects over 90% of Parkinson's disease patients. Recently, subthalamic nucleus deep brain stimulation has shown promise for alleviating sleep dysfunction. We previously showed that a single-layer neural network could classify sleep stages from local field potential recordings in Parkinson's disease patients. However, it was unable to categorise non-rapid eye movement into its different sub-stages. Here we employ a larger hidden layer network architecture to distinguish the substages of non-rapid eye movement with reasonable accuracy, up to 88% for the lightest substage and 92% for deeper substages. Using Shapley attribution analysis on local field potential frequency bands, we show that low gamma and high beta are more important to model decisions than other frequency bands. These results suggest that the proposed neural network-based classifier can be employed for deep brain stimulation treatment in commercially available devices with lower local field potential sampling frequencies.

[1] Department of Medical Biophysics, University of Toronto, Ontario, Canada. [2] Department of Biomedical Engineering, McGill University, Montreal, Canada. [3] Department of Anesthesiology, University of Colorado Anschutz Medical Campus, Aurora, CO, USA. [4] Department of Neurosurgery, University of Nebraska Medical Center, Omaha, NE, USA. [5] Department of Physics and Astronomy, York University, Ontario, Canada. [6] Department of Neurology, University of Colorado Anschutz Medical Campus, Aurora, CO, USA. [7] Department of Neurosurgery, University of Colorado Anschutz Medical Campus, Aurora, CO, USA. [8] Department of Psychiatry, University of Colorado Anschutz Medical Campus, Aurora, CO, USA. [9] These authors jointly supervised this work: Aviva Abosch, Joel Zylberberg, John A. Thompson. ✉email: john.a.thompson@cuanschutz.edu

Deep brain stimulation (DBS) for Parkinson's disease (PD) is a well-established, highly effective therapy for motor symptoms[1]. Over the last 20 years, observational and experimental studies suggest that subthalamic nucleus (STN; one of the two FDA approved brain targets for treating the motor symptoms of PD) stimulation via DBS also might ameliorate sleep dysfunction[2–6]. Though sleep deficits in PD are highly prevalent and negatively impact quality of life there currently exist no direct treatments addressing sleep deficits[7–9]. Prospective modification of sleep stage expression (i.e., time in rapid-eye-movement (REM) sleep) requires synchronized modulation of sleep-stage-dependent brain activity. Direct sensing of local field potential (LFP) data from the DBS electrodes has recently become commercially available, enabling the inference of sleep stage from deep brain signals[10], which builds from established work in ECoG (electrocorticography), EEG (electroencephalography) and acute DBS recordings[11–14]. In principle, such inference can then be used to direct closed loop DBS modulation to specific sleep stages to improve sleep.

Our prior work shows LFP signals are sufficient to accurately categorize 30-second epochs of sleep recordings as REM, awake, or non-rapid eye movement (NREM) sleep[15] (Fig. 1a, b). Importantly, this previous work did not attempt to sub-categorize the three NREM stages and instead combined them into a single category label. This lack of sub-categorization precludes the tailoring of DBS stimulation settings to specific NREM subtypes (e.g., NREM1). This limitation is important because REM and NREM3 in particular are less prevalent in PD patients vs age-matched controls, and these sleep stages constitute *restorative* sleep. To overcome this limitation, the current study's primary aim was to develop new neural network (NN) algorithms capable of sub-categorizing NREM sleep. Our secondary aim was to understand whether sleep stage categorization can be accomplished using LFP data gathered at lower sampling rates. While the dataset studied in our previous work was collected at 1024 Hz, current commercially available DBS platforms record LFP signals at ranges of 250 to 1000 Hz[15,16] (Fig. 1a). For this study, we compared our original model to a downsampled model of 250 Hz for two reasons: firstly, to determine if our approach would work on the broadest range of DBS devices currently commercially available, and secondly, to determine how much classification performance benefits from the use of higher sampling rates.

In addition to basal ganglia recordings in PD, automatic classification of sleep stages from intracranial recordings, both cortical and hippocampal, have been explored in epilepsy[10]. In general, machine learning methods that have been applied to sleep stage classification from intracranial data include support vector machines (SVMs), hierarchical clustering with decision trees, or combinations of these methods[10,14,17,18]. For input data, most published methods use power spectral data from frequency band bins (e.g., beta: 13–30 Hz)[10,14,17]. In addition to targeted sleep stage classification, recent efforts in PD and epilepsy have observed multi-scale biorhythms that could be used to inform adaptive stimulation control schemes[19]. Although our current work is predicated on single-stage resolution for intervention, one or more diurnal neural biorhythms could be explored as an adjunctive strategy.

In earlier work, supervised classification approaches performed poorly in differentiating NREM states or awake from REM in neural data either derived from scalp or intracranial recordings. In our prior work, combining the NREM stages enabled an improvement in overall categorization performance, with a 91% prediction accuracy averaged across all labels[15]. The reduced expression of NREM3 in our subjects motivated a composite classification grouping of NREM1-3. The key strategies enabling our previous improvement in categorization performance were

(1) the normalization of LFP signal amplitudes before inputting them into the classifier, and (2) the use of NNs in place of weaker classifiers such as SVMs[14], and (3) inverse-frequency weighting in the loss function to account for increased class sampling imbalance (e.g., smaller training rewards for batched common sleep stages). At the same time, our previous NN approach had several limitations, including lower prediction accuracy for REM stages and poor performance in distinguishing between the different NREM states. Following exploration of other machine learning approaches, we opted to use an NN approach for the advantages associated with better handling of complex data and greater end-to-end learning and flexibility.

To overcome the limitations of our prior approach, here we expand upon our previous model by implementing larger NNs and training them for much longer. This approach leverages the recently identified double descent phenomenon[20]: As model size or training time increases, NN models may initially overfit to their training data, but continuing to increase the model size or training time further typically alleviates the overfitting. This observation means that, in general, the best model size and training duration are the largest ones possible within the computational resources available. With the larger size of our new model, we more efficiently divide NREM into sub-groups of NREM1 and a combined NREM2&3 label. Given the well-known shortage of NREM3 in this clinical population, it is typically excluded from model development[14,17]. Nevertheless, the advances we report here enable DBS stimulation settings to be separately tailored to NREM1 versus NREM2&3, which is expected to improve the ability of adaptive stimulation to ameliorate sleep deficits[10]. This effort builds on past work that has demonstrated the feasibility of using DBS LFP signals for sleep classification in which an embedded linear classifier to determine asleep and awake states[21].

Finally, we sought to assess the relationship between two frequency-related aspects of the data and sleep stage prediction: (1) LFP frequency band contribution and (2) sampling frequency of the data. SHAP (SHapley Additive exPlanations) analysis (Fig. 2) was applied to LFP frequency bands and demonstrated that low gamma and high beta contributed most to the model prediction. With regard to sampling frequency, we tested our larger models on LFP data that were downsampled to match the lowest available sampling frequency of current commercial DBS devices[16] (Fig. 2). We found that performance improvements over our previously published small NN persisted. Altogether, this work suggests that by leveraging double descent, we can build sleep-stage classifiers that will work with data gathered at lower sampling frequencies and will differentiate between the different NREM stages. Collectively, these advances improve the prospects for developing adaptive DBS treatments tailored to target specific sleep stages, thereby mitigating the sleep dysfunction suffered by PD patients. On-going efforts to selectively target specific sleep states for external[22,23] and internal[24] stimulation to improve overall sleep quality will benefit from this work. With regards to PD, stage-targeted stimulation may improve NREM duration and expression and secondarily minimize WASO (wake after sleep onset), which is implicated in restorative rest and overall sleep maintenance and quality[22,24].

## Methods

**Patient demographics**. This study was approved by the Institutional Review Board of the University of Minnesota, where the surgical and recording procedures were performed. All consenting study subjects ($n = 10$) carried a diagnosis of idiopathic PD (Table 1 & Fig. 1a, b). Subjects were unilaterally implanted in the STN with a quadripolar DBS electrode (model #3389: Medtronic

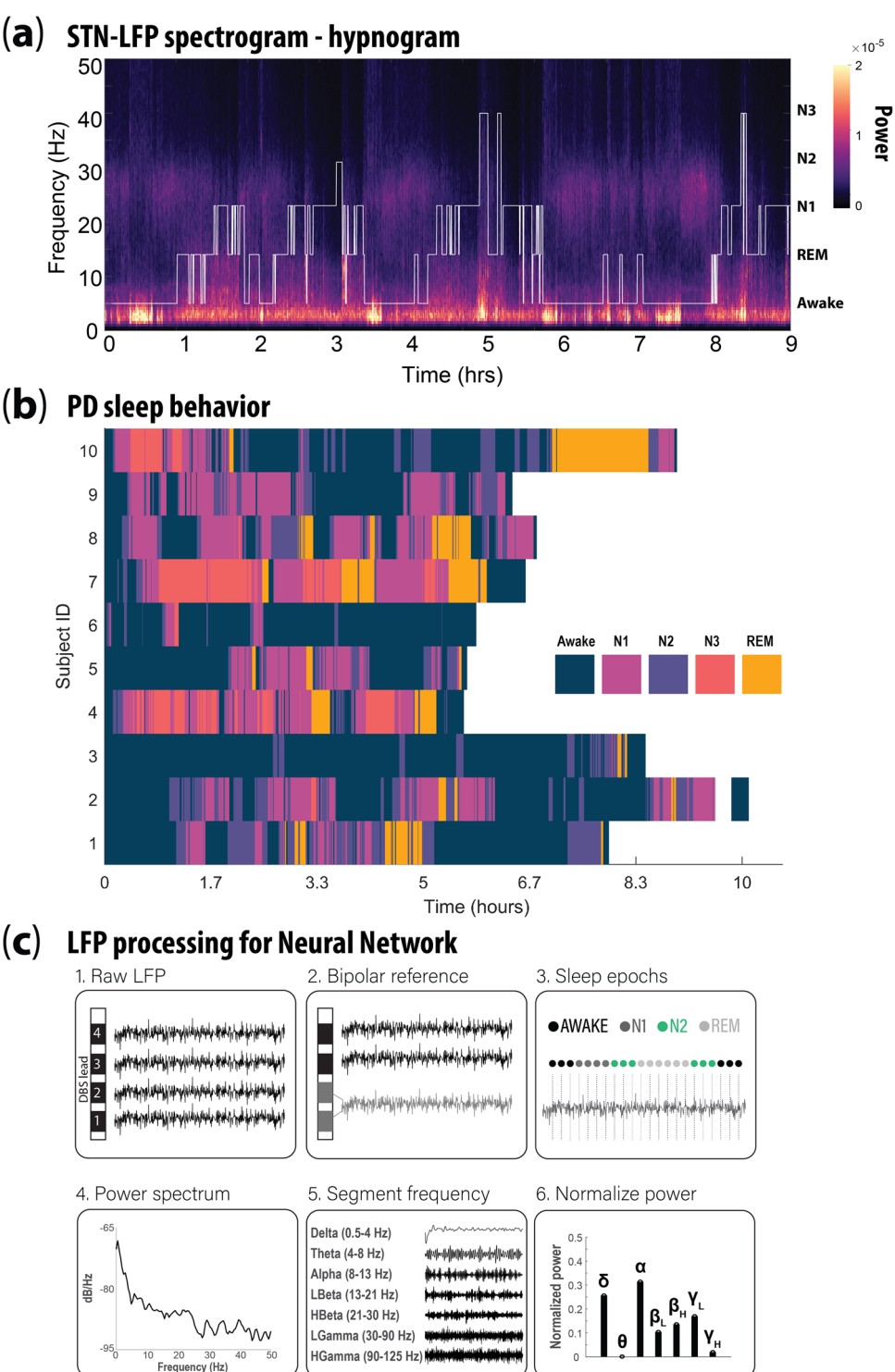

**Fig. 1 Overview of local field potential recordings collected during sleep in Parkinson's disease patients. a** Representative local field potential (LFP) spectrogram recorded from the subthalamic nucleus (STN) of a Parkinson's disease (PD) subject during a single nocturnal sleep period. White line indicates the hypnogram. Colorbar denotes the local field potential power (LFP) magnitude. **b** A horizontal stacked bar chart depicting the occurrence and distribution of sleep stages (i.e., awake, NREM1, NREM2, NREM3 and REM) for all 10 PD subjects used in this study. Colors denote distinct sleep stages. **c** Preprocessing steps for LFP data used as input for NN sleep stage classification (LFP data in schematic were simulated and not subject data). (1) Individual LFP recordings from each deep brain stimulation contact (black squares = DBS recording contact), (2) bipolar referencing strategy; grey squares indicate two adjacent DBS contacts used for referencing and resulting grey LFP recording, (3) raw LFP is partitioned into 30 s epochs based on distinct sleep stages (defined by expert scored polysomnography), (4) time and frequency processing performed on each 30 s epoch of LFP, (5) each 30 second epoch of LFP was decomposed into canonical frequency band related information, (6) power contributions for each frequency band were normalized before input into neural network.

## (a) Artificial Neural Network

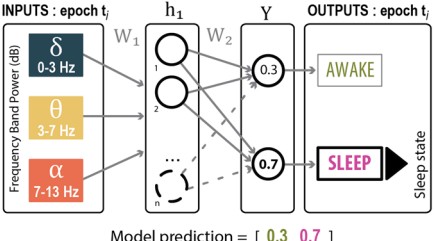

Model prediction = [ 0.3  0.7 ]

## (b) Feature Coalition Probe

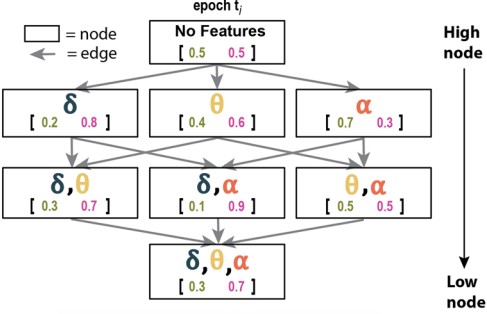

## (c) Feature Contribution

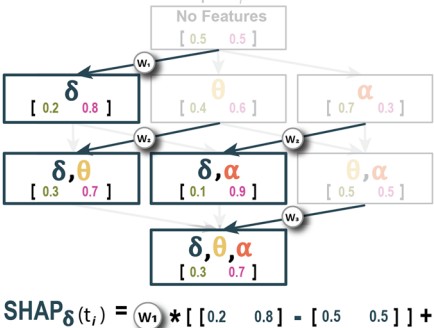

$$\text{SHAP}_\delta (t_i) = W_1 * [\ [\ 0.2 \quad 0.8\ ] - [\ 0.5 \quad 0.5\ ]\ ] +$$
$$W_2 * [\ [\ 0.3 \quad 0.7\ ] - [\ 0.4 \quad 0.6\ ]\ ] +$$
$$W_2 * [\ [\ 0.1 \quad 0.9\ ] - [\ 0.7 \quad 0.3\ ]\ ] +$$
$$W_3 * [\ [\ 0.3 \quad 0.7\ ] - [\ 0.5 \quad 0.5\ ]\ ]$$

$$\text{Where } W_n = \frac{1}{\left(n \cdot \binom{f}{n}\right)}$$

$n$ = number of features included in that node,
and $f$ = total number of features = 3 (delta, theta, alpha)

## (d) SHAP Vector

$$\text{SHAP}_\delta = \begin{bmatrix} \text{SHAP}_\delta (t_0) \\ \vdots \\ \text{SHAP}_\delta (t_i) \\ \vdots \\ \text{SHAP}_\delta (t_{N-1}) \end{bmatrix}$$

Where $N$ = the total number of epochs

**Fig. 2 SHapley Additive exPlanations (SHAP) analysis for frequency band contributions to model prediction. a** Two-state (awake, sleep) array prediction by a NN using three input feature bands (delta, theta, alpha) at epoch $t_j$. **b** The power set of all features, i.e. all unique subsets of features. Each node contains the model prediction if only the features within that node are known to the model. **c** An example of the calculation of the marginal contributions of the delta band at epoch $t_j$ to the model prediction, and the SHAP equation for the delta band at that epoch. This is the weighted sum of all the marginal contributions. **d** The complete SHAP vector for the delta band.

### Table 1 Summary of model performance metrics

| Class | Accuracy | Sensitivity | Specificity |
|---|---|---|---|
| 3 Classes Original Neural Network (NN) | | | |
| awake | 0.93 | 0.92 | 0.95 |
| NREM | 0.92 | 0.95 | 0.9 |
| REM | 0.96 | 0.46 | 0.99 |
| 3 Classes Large NN | | | |
| awake | 0.96 | 0.96 | 0.96 |
| NREM | 0.94 | 0.94 | 0.95 |
| REM | 0.97 | 0.66 | 0.98 |
| 4 Classes Original NN | | | |
| awake | 0.94 | 0.96 | 0.92 |
| NREM2&3 | 0.9 | 0.89 | 0.91 |
| NREM1 | 0.86 | 0.48 | 0.93 |
| REM | 0.96 | 0.47 | 0.99 |
| 4 Classes Large NN | | | |
| awake | 0.96 | 0.97 | 0.95 |
| NREM2&3 | 0.92 | 0.88 | 0.94 |
| NREM1 | 0.88 | 0.59 | 0.94 |
| REM | 0.97 | 0.75 | 0.99 |
| 4 Classes Large DS (250 Hz) NN | | | |
| awake | 0.94 | 0.93 | 0.95 |
| NREM2&3 | 0.9 | 0.88 | 0.9 |
| NREM1 | 0.86 | 0.52 | 0.93 |
| REM | 0.96 | 0.62 | 0.98 |

**Signal processing**. Signal processing of the raw STN LFP signals has been previously described[14,15] Fig. 1c. Externalized lead recordings were collected at 1024 kHz. Briefly, after preprocessing, the four LFP channels were converted into three bipolar derivations, using sequential contacts as a reference (e.g., LFP01, LFP12 and LFP23). Power spectral density (PSD) was estimated using a fast Fourier transform from a 2-second-long sliding Hamming window with 1-second overlap. The final time-evolving spectra had 15-second time and 0.5 Hz frequency resolution. For each subject, LFP data selected for further analysis were based on which DBS lead contact(s) within the STN were associated with peak beta-spectrum activity, as this feature correlates with the optimal programming contact(s) for the treatment of contralateral motor symptoms[26]. We compared a model with two hidden layers and 1000 units per hidden layer to our previously published model architecture, which was composed of a single hidden layer containing 32 units. We refer to the new, larger model as the 'large' NN, and refer to our previously published small architecture as the 'original' network. The inputs for both the original model (Figs. 3a, c and 4a, d) and the large model (Figs. 3c, d and 4b, d) consisted of eight separate frequency band power bins, averaged over 30 s: delta (0–3 Hz), theta (3–7 Hz), alpha (7–13 Hz), low beta (13–20 Hz), high beta (20–30 Hz), low gamma (30–90 Hz), high gamma (90–200 Hz) and high frequency oscillations (200–350 Hz). Each frequency range input feature was normalized independently by subtracting the mean

Inc., Fridley, MN), per routine surgical protocol[25]. Experimental details for the recording setup have been previously published[14]. A basic characterization of these data was previously reported in ref. [14] and initial model development was described in Christensen et al.[15].

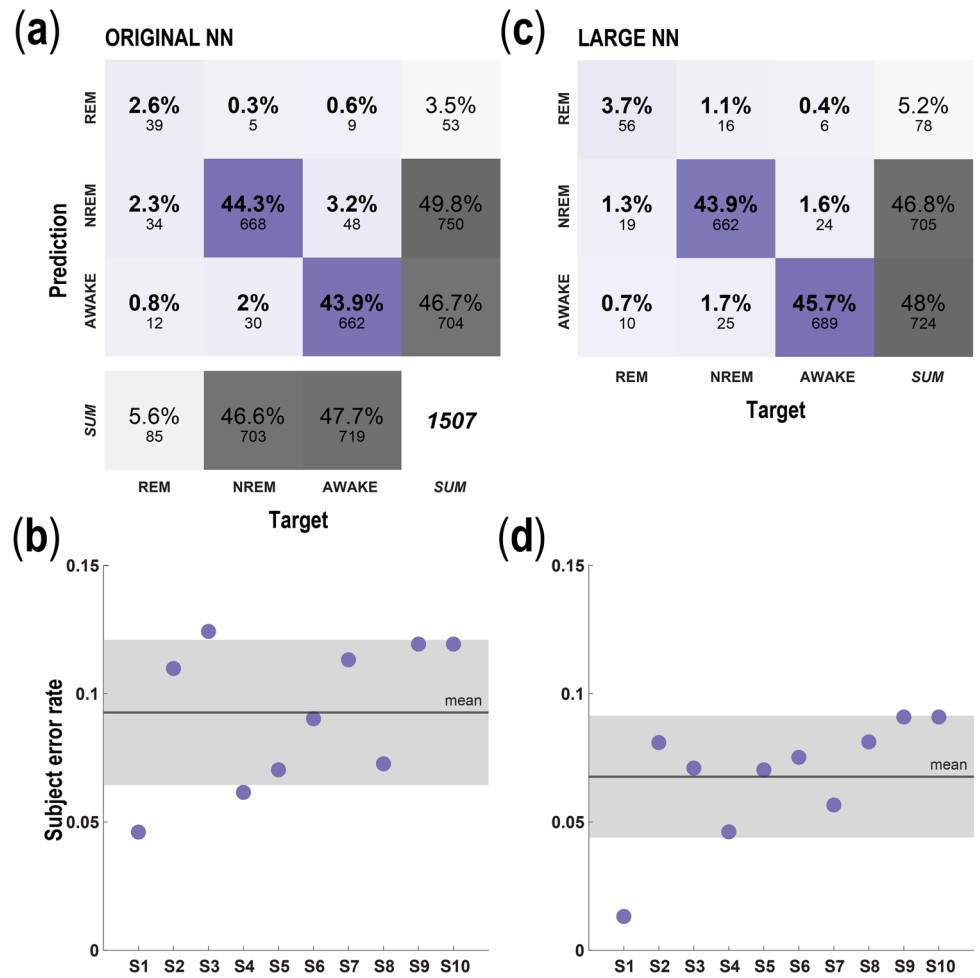

**Fig. 3 Original model performance. a** Confusion matrix representation of the prediction accuracy for the original single hidden layer, 32 node NN (same architecture as Christensen et al. 2019); color (purple) intensity within the stage comparisons indicates percent whereas gray indicates percent density for the sum across categories. **b** Average error rate of the network from A, when tested on data from each of the 10 subjects. Gray box indicates standard deviation. **c** Same as in **a**, depicting the accuracy for the new Large NN (two layers, 1000 nodes per hidden layer). **d** Same as in **b**, scatterplot of the error-rate for each subject using the large NN. Note the substantial reduction in error rate with the large model. All results in this figure were obtained using held-out test data.

and scaling by the variance of the feature, with scaling performed separately for each patient. For the downsampled model (Fig. 4e, f), the raw voltage data were downsampled to 250 Hz before preprocessing: after this downsampling, the same preprocessing was applied to these downsampled data as to the original data. In the case of the downsampled data, the highest frequency band was absent, and the model inputs thus included only seven frequency bands: delta (0–3 Hz), theta (3–7 Hz), alpha (7–13 Hz), low beta (13–20 Hz), high beta (20–30 Hz), low gamma (30–90 Hz), and high gamma (90–125 Hz).

**Polysomnography scoring.** LFP and polysomnography (PSG) were concurrently recorded in all subjects. Briefly, we used the AASM-recommended (PSG) electrode montage that included the following: F3–C3, P3–O1, F4–C4 and P4–O2, EOGL–A2, EOGR–A1, and chin EMG[27]. Sleep stages were determined by analysis of 30-s epochs of the PSG by a sleep staging expert, with each epoch classified as awake or as belonging to one of the following four sleep stages: REM, NREM1, NREM2, or NREM3. Our recent consensus efforts sought to improve the accuracy of sleep stage classification by expert reviewers in the context of patients with PD comorbid with sleep dysfunction[28]. Expert-derived sleep stages, based on polysomnography evaluation, were

used to calculate the total duration of sleep over the period of one night. To estimate the total duration of sleep, scored epochs were summed excluding periods of wakefulness, from the time the individual first fell asleep to the time of final awakening; for details, please see Fig. 1 from ref. [15].

**Model descriptions and hyperparameter optimization.** Similar to our previous study[15], we trained feedforward NNs to take in the vectors of LFP power at each epoch, and to output the network's inferred probability associated with each sleep state label. For the results in Fig. 3, these outputs were 3-dimensional vectors with entries associated with the following categories: awake, REM, and NREM, where all three NREM states were combined into one label. For the results in Fig. 4, the outputs were 4-dimensional vectors, with entries associated with the following categories: awake, REM, NREM1, and a label combining NREM2 and 3 (i.e., NREM2&3). In this case, NREM stages 2 and 3 were combined into one label because of the minimal expression of NREM stage 3 to inform the model. The models were trained on 75% of the epochs from the dataset and were tested on the remaining held-out 25%. All networks were trained to minimize the categorical cross-entropy loss between their predictions and the PSG-scored sleep state labels.

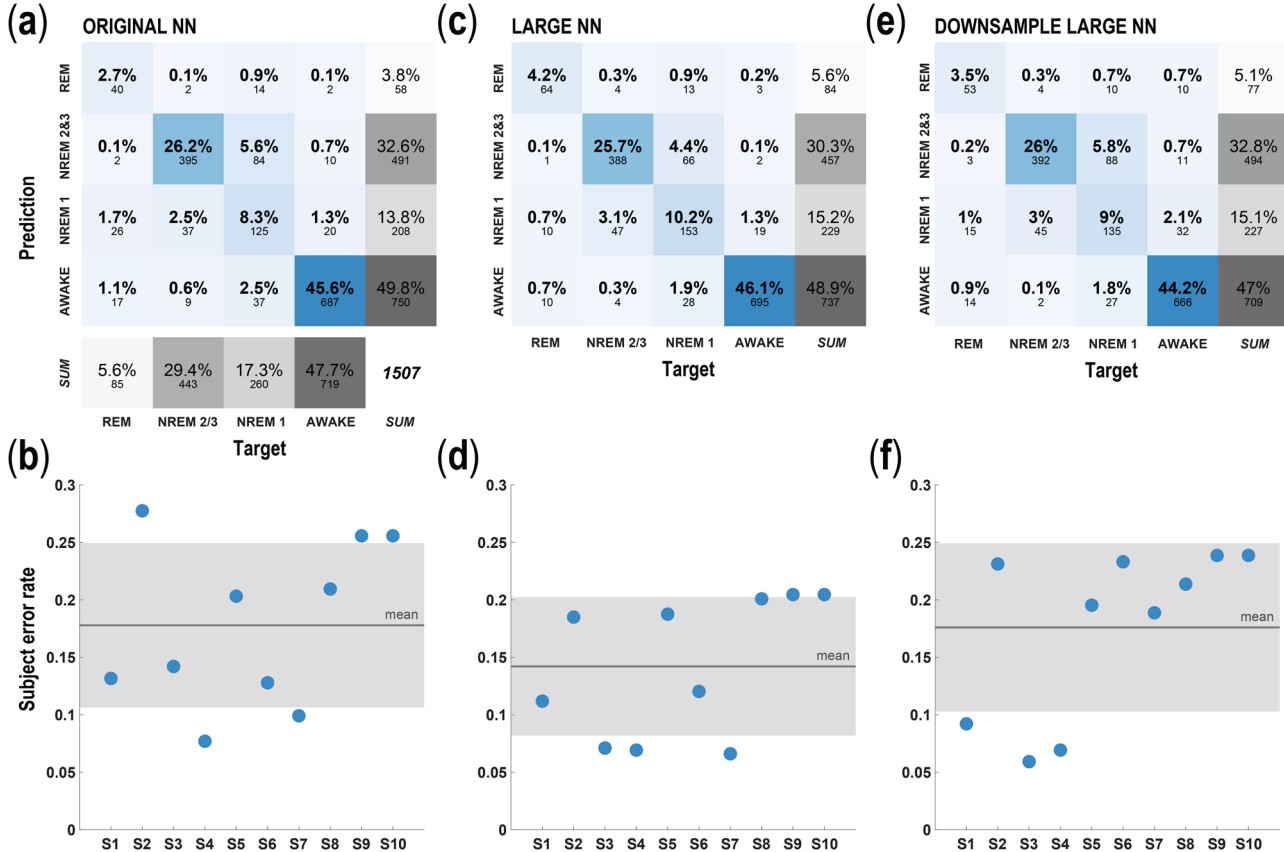

**Fig. 4 Large model performance. a** Confusion matrix representation of the prediction accuracy for the original single hidden layer, 32 node NN (Christensen et al. 2019) using four sleep stage labels (i.e., awake, NREM1, NREM2&3, REM); color (blue) intensity within the stage comparisons indicates percent whereas gray indicates percent density for the sum across categories. **b** Average error rate of the network from **a**, applied to the four sleep stage labels for each of the 10 subjects. Gray box indicates standard deviation. **c** Similar to **a**, depicting the accuracy for the new large NN. **d** Similar to **b**, scatterplot of the error rate for each subject using the large NN and high temporal frequency data. **e** Same as in **a**, **c**, showing the accuracy of the large NN using downsampled (250 Hz) LFP data. Downsampling the LFP data using parameter limits imposed by available sensing DBS implanted programmable generator (IPG) technology (250 Hz) somewhat reduced prediction accuracy for the four sleep-stage labels using the large model. **f** Same as in **b**, **d**, scatterplot of the error rate for each subject using the large NN applied to the downsampled data. All results in this figure were obtained using held-out test data.

For the original NN models (Fig. 3a, b, and Fig. 4a, b), we used the same architecture as in Christensen et al.[15], consisting of a densely connected NN comprised of a single hidden layer with 32 rectified linear (ReLu) units, followed by the output layer consisting of three or four units (depending on whether or not NREM stages were subdivided) and a softmax nonlinearity. These models were trained for 100 epochs using the stochastic gradient descent (SGD) optimizer and a fixed learning rate of 1. Consistent with the findings of Christensen et al.[15], the performance of these smaller models was not strongly dependent on their hyperparameters, such that increasing the number of hidden units by a factor of 2, and/or decreasing the learning rate by a factor of 10 did not demonstrably change the model accuracy.

For the large NN models (Fig. 3c, d, and Fig. 4c–f), we used feedforward NNs with two hidden layers, each containing 1000 ReLu units, followed by the output layer consisting of three or four units, depending on how many sleep states were used in the experiment. Batch normalization (*batchnorm*) was included after each hidden layer, as was dropout regularization, with a drop probability of 50%. These large models were trained for a total of 1000 epochs using the SGD optimizer. Their initial learning rate was 0.1, and the learning rate was reduced by a factor of 2 after every 100 epochs. A limited hyperparameter search was performed, which included the addition of another hidden layer, and reducing or increasing the width of the hidden layers by a

factor of 2: These changes did not substantially affect model accuracy. However, removing either dropout or *batchnorm*, or making the learning rate constant, did demonstrably reduce accuracy.

**SHapley Additive exPlanations (SHAP) analysis**. To understand the relative importance of each of the different LFP frequency input bands for classifying sleep stages, we computed SHAP (SHapley Additive exPlanation) values[29,30]. In brief, for each epoch we used SHAP analysis to determine the marginal contribution of each LFP band to model output, then combined the SHAP values over all epochs to measure how each feature contributed to the model overall. Figure 2 represents a simplified version of our approach. Figure 2a illustrates a two-state (awake, sleep) array prediction by an NN using three input feature bands (delta, theta, alpha) with model predictions of 0.3 and 0.7 for each state, respectively (Fig. 2a). To calculate the SHAP values for epoch $t_j$, we first generate all the unique subsets of features, then for each subset determine the prediction of the model at that epoch if only the features within that subset are known to it. In the 'no features' case, the prediction of the model simply reflects the presence of that state in the data (Fig. 2b). The marginal contribution is calculated for each feature by finding the difference in prediction arrays between lower nodes where the feature is present and higher nodes where the feature is not, where the

two nodes are connected by an edge. The SHAP value array for that feature and epoch is the weighted sum of these contributions and has the shape 1× num(states) (Fig. 2c). Therefore, the SHAP value for an epoch is equal to the weighted sum of all the differences in model outputs when considering all possible subsets (coalitions) of input features. The complete SHAP array for one feature (e.g., delta band) consists of all the SHAP feature vectors over all epochs and has shape num(epochs) × num(states) (Fig. 2d). The SHAP values were computed for each input LFP band, for each 30-s epoch of the testing dataset, and for the set of all considered sleep stages. Figure 5a–d shows the median of the absolute values of the SHAP arrays for all epochs and input LFP bands. These were computed for the large NN, consisting of two hidden layers and 1000 hidden units per layer on the four-state classification problem.

## Results

**Comparison of three sleep-stage label prediction between single layer vs. large NN.** We sought to develop a model with increased accuracy for predicting sleep stage intracranial LFP recordings from a chronically implanted STN DBS lead. We first studied the task of classifying LFP signals into three different categories (awake, REM, NREM), and compared a model with two hidden layers and 1000 units per hidden layer to our previously published model architecture, which was composed of a single hidden layer containing 32 units.

We compared the error rates of these three-state models (error rate = incorrect/(correct + incorrect)) and observed that the large network had an error rate of 6.7%, while the original network had an error rate of 9.2% (Fig. 3c, d). In other words, the large model obtained the correct categorization—and hence avoided the errors made by the original model—on roughly 27% of the data samples for which the original model architecture returned the incorrect classification. NREM state predictions were slightly overrepresented in the original NN; but were more closely aligned with the PSG-derived labels in the large architecture. For both models, the performance values varied somewhat across subjects (Fig. 3b, d).

**Comparison of four sleep-stage label predictions between single layer versus large NN.** In addition to developing a more robust model, we sought to increase the capacity of our model to discriminate between NREM sleep stages, which our prior model did not accommodate (Fig. 3a, b). Given the extreme shortage of NREM3 epochs in our dataset—a feature of sleep dysfunction in PD—we were still unable to distinguish between NREM2 and NREM3 in preliminary testing. We compared two updated models, both predicated on separating NREM sleep stage epochs into NREM1 and combined NREM 2 and NREM 3. (Christensen et al., 2019)[15]. We observed that the large network performed better than the original network. Specifically, the large network with NREM differentiation had a lower error rate of 13.7% compared to the original network architecture with NREM differentiation (17.3%; Fig. 4b, d). This means that, given the same target of classifying between 4 sleep states, roughly 20% of the errors made by the original network were avoided by the large model. Both architectures confused NREM1 and NREM2&3 for each other most often. The next most common mistake was misclassifying NREM1 as awake. As with the models trained to predict 3 sleep stages, the larger model with more fine-grained sleep stage differentiation performed better at classifying REM states than the original architecture. NREM2&3 was the state most often erroneously classified as REM sleep. The original architecture overestimated NREM2&3 as compared to NREM1, however, model performance varied somewhat between subjects

(Fig. 4b, d). REM states were underrepresented as predictions in the original NN, which was a deficit ameliorated in the large NN. We found that the large NN predicted more REM overall and exhibited more accurate classification of REM (66% correct REM classification in the large network vs 45% correct REM classification in original).

**Impact of downsampling on four-sleep-stage label prediction.** The models described above were trained and tested using LFP data recorded at 1024 Hz, which included the high gamma (90–200 Hz) and the high frequency oscillations (HFO, 200–350 Hz). However, currently available FDA-approved DBS implanted programmable generators (IPGs) that permit recording from brain leads are limited to 250 Hz sampling resolution. Given the Nyquist limit, this means that currently available IPGs capable of recording can resolve LFP signals up to 125 Hz. Furthermore, while other DBS devices with higher sampling resolutions exist (e.g., Medtronic Summit RC + S system), we again focus on the most limited resolution case to understand how much sleep-stage classification performance is degraded in cases where sampling resolution is limited.

To determine how much this limitation reduces our ability to infer sleep stages from LFP signals, we downsampled our LFP data to 250 Hz. This resulted in the exclusion of the HFO band, and a reduction in the width of the high gamma band, from 90–200 Hz down to 90–125 Hz. Figure 4e, f shows that downsampling the data to match current recording constraints does somewhat increase the model error rate (overall error rate 13.7% for high temporal resolution data versus 17.3% for downsampled data). This means that roughly 25% of the errors made from models constrained to 125 Hz LFP signals are avoided with the inclusion of higher frequency data. However, the large model evaluated on the downsampled data, and the original model evaluated on the full-resolution data had similar overall error rates. To further refine this analysis, we created additional models for data sampled at 125 Hz, 400 Hz, and 700 Hz. Figure 6b demonstrates that there is minimal degradation of model performance based on sampling resolution.

**Attribution analysis using SHapley Additive exPlanations (SHAP) analysis.** To further understand the contribution of each LFP frequency band to each sleep stage prediction, we conducted a SHAP (SHapley Additive exPlanations) analysis. SHAP analysis uses Shapley values computed from coalitional game theory to calculate the individual contribution of each frequency band to the model's categorization of each sleep stage for each epoch (see Methods). We performed this analysis on the large network that was trained to classify the high-resolution LFP inputs into four different sleep stages: awake, REM, NREM1, and NREM2&3.

Low gamma contributed most to the model prediction for the awake state (Fig. 5a); high beta contributed most to the model prediction for NREM1, combined NREM2&3, and REM (Fig. 5b–d). Figure 5e depicts the relative contributions of the LFP bands, highlighting that high beta, low gamma, and high gamma contributed most to predictions on average. The high gamma frequency band was only in the top three contributing features for the awake label; this is consequential as high gamma is reduced in currently available FDA-approved implantable pulse generators, which can only capture signals up to 125 Hz. The secondary importance of the high gamma band and the relative unimportance of the low-frequency bands such as delta and alpha to sleep stage categorization suggest that a model using an even greater constraint of feature bands may be possible while retaining accuracy. Here, we classify high gamma signals up to 200 Hz (and HFOs up to 350 Hz) in our full resolution data.

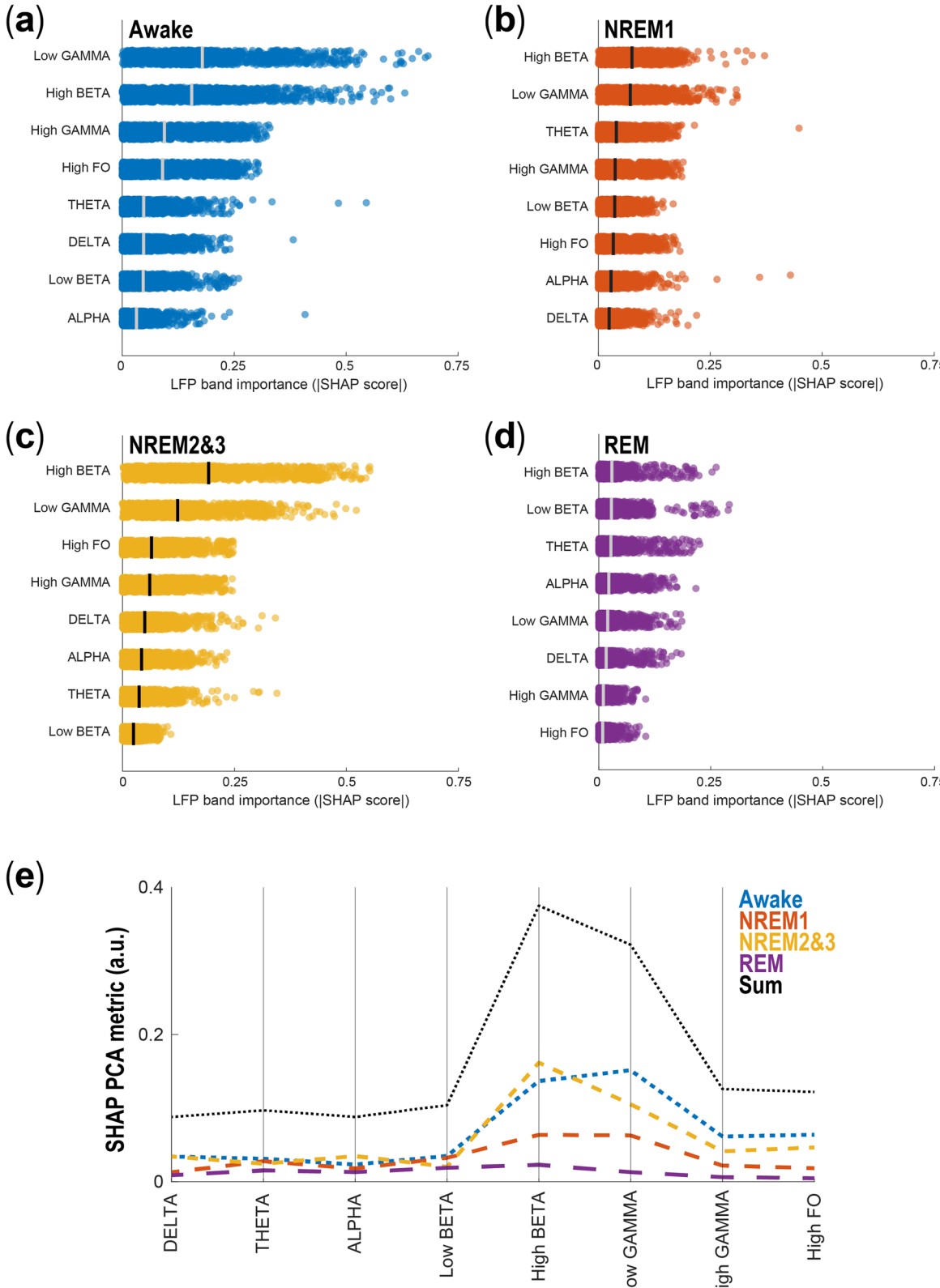

**Fig. 5 SHAP derived LFP frequency band contributions. a** Column scatter plot depicting the LFP band importance (SHAP score) for awake epochs. **b** Same as in **a**, for NREM1 epochs, **c** Same as in **a**, for NREM2&3 epochs, **d** Same as in **a**, for REM epochs. **e** Parallel coordinate plot showing the median of the SHAP array for that specific state and epoch, with multidimensional scaling to permit comparison of the relative magnitude that specific LFP bands contribute to the categorization of each sleep stage.

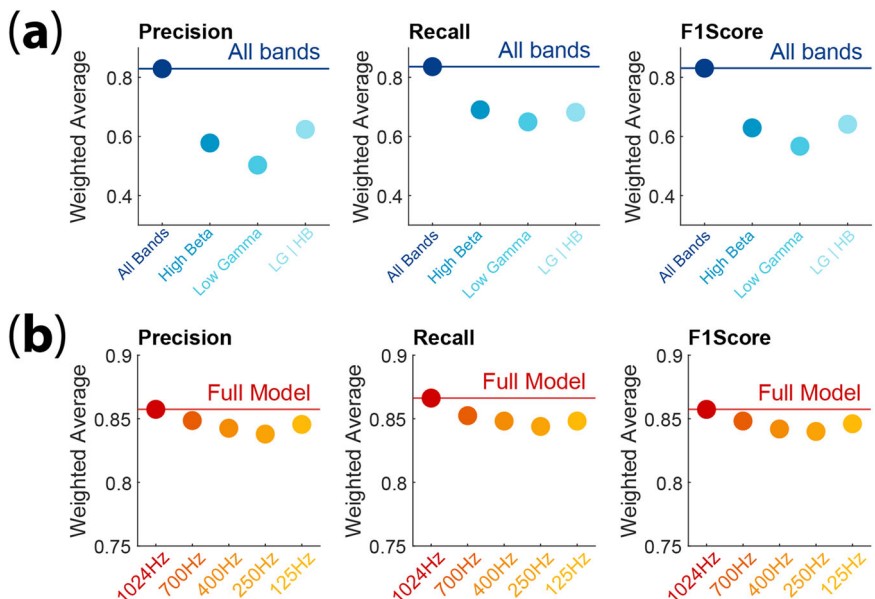

**Fig. 6 Evaluation of sub-band and sampling frequency parameters on model performance. a** Performance metrics for separate neural network (NN) models trained on sub-bands of the total frequency range used in the full model (i.e., high-beta (HB), low-gamma (LG), and high-beta plus low gamma (HB/ LG)). These sub-bands were selected based on their contribution to the full model evaluated using SHAP analysis. Models based on sub-bands performed poorly compared to the model based on all bands. The following performance metrics were evaluated: precision, porportion of correctly predicted positive observations compared to the total number of instances predicted as positive; recall, proportion of correctly predicted positive observations to all correctly identified observations (true positives and false negatives); and F1score, the harmonic mean of precision and recall. **b** Performance metrics for separate NN models trained on varying sampling resolutions including all bands before downsampling (i.e., 125 Hz, 250 Hz, 400 Hz, 700 Hz, and the full model–1024 Hz). Lower resolution data did not markedly affect model performance.

The above findings help explain the increase in categorization errors we observed when we attempted to classify sleep stages using temporally downsampled data (Fig. 4e, f). Specifically, because high gamma and HFO bands contribute somewhat to the categorization (especially for awake and NREM2&3 sleep stages, which are most common in the dataset), their absence causes a reduction in model prediction accuracy. Nevertheless, current IPGs do fully capture high beta (21–30 Hz) and low gamma (30–90 Hz) frequency bands, which SHAP analyses flag as the most important frequency bands for classifying awake and NREM sleep states. This can explain how sleep stage categorization is still possible, albeit with reduction in accuracy, even when using temporally downsampled data.

To assess the predominant contribution indicated by SHAP analysis of high beta and low gamma, three additional models were generated using either high beta alone, low gamma alone, or the combination of high beta and low gamma. In Fig. 6a, we observed that all three of these models performed poorly compared to the model including all bands for several performance metrics (recall, precision, and F1-score). This indicated that although specific frequency bands appear to contribute more than others to predictive performance on specific sleep stage classes, the model utilizes information from all frequency bands.

### Discussion
We sought to refine a set of algorithms that classify sleep stages using LFP signals recorded by electrodes implanted in the STN of PD patients. Algorithms that can perform this task enable adaptive DBS which can be used to target specific sleep stages and help reduce sleep deficits suffered by PD patients. In this study, leveraging double descent[20], we used larger NNs as well as longer training regimes than we had previously studied[15]. These larger networks were able to sub-classify NREM sleep into NREM

1 vs. NREM2&3 better than the original networks from our previous work. Our work suggests that larger NNs will be useful in developing adaptive DBS treatments. Notably, even in the case of hardware limitations, it may be possible to use model distillation methods[31] to make smaller models that can mimic our large NN at lower computational cost, although we leave such efforts for a future study.

In addition to the question of NREM sub-classification - a challenge posed by electrophysiological signal ambiguity, model complexity, and limited expression of NREM3 in PD patients- we also addressed the practical question of whether LFP-based sleep stage categorization could succeed with existing commercially available hardware and sampling constraints (i.e., the 250-Hz limit). Compared with higher-resolution LFP data (1024 Hz) collected with a clinical neurophysiological recording system, we saw a moderate loss of classification performance when the lower-resolution LFP data were used for the classification. While our results suggest that adaptive DBS treatments could potentially work with lower temporal resolution LFP recordings, they also indicate that these treatments could benefit from increasing the temporal resolution of the LFP sampling afforded by the IPG. Specifically, classification performance was reasonable even with downsampled LFP data, but was substantially higher when a larger range of input frequencies was used.

It is important to note that our training set consisted of data annotated by a single human sleep expert. Independent experts scoring the same data typically achieve approximately 76.8-82% consensus for sleep stage classification[32,33]. This level of agreement could arise from inherent ambiguity regarding some of the sleep stages (i.e., from uncertainty about the ontology of sleep). The level of inter-rater reliability sets a soft ceiling on how well we might expect classification algorithms to perform. Outside of the current study, we have recently developed a consensus-based scoring method in which multiple experts individually annotate

each night's sleep, and then reach consensus on epochs of disagreement through collective review of disputed PSG epoch and synchronous video data[28]. We anticipate that these efforts will help to clarify the true upper bound on classification performance that is attainable within the context of current sleep stage ontology. With regard to NREM subclassification, while our large new NN model could successfully divide NREM sleep into two sub-categories (NREM1 and NREM2&3 combined), it could not distinguish between NREM stages 2 and 3. This limitation arises from the dearth of NREM3 epochs within our dataset, which is a known characteristic of sleep in individuals with PD[17], and has also stymied previous efforts to sub-categorize NREM sleep in PD patients.

Consistent with many previous studies[10,14,15,17], we preprocessed the LFP signals by computing the power in different frequency bands as inputs for classification (Fig. 1c). At the same time, one of the key messages of the deep learning revolution is that extrapolated data features (e.g., LFP frequency bands), as opposed to the raw signal(s), often underperform relative to end-to-end training schemes in which the relevant features are optimized for the classification task[34]. This suggests that we may be able to further improve our classification performance by using the raw LFP time series as inputs. Preliminary investigations using NNs composed of long short-term memory (LSTM) failed to yield any performance gains relative to the feedforward networks presented here. While more study is required, this could indicate that model training on a larger dataset is critical. Future studies aiming to advance this goal could benefit from foundation model approaches[35], involving the training of models using unsupervised predictive loss functions on larger reams of potentially unlabeled LFP data from both PD patients and other populations (including non-PD subjects, for instance those with epilepsy or obsessive-compulsive disorder[10]). We define a foundation model as a model that is first trained in a self-supervised manner using large quantities of unlabeled data, then is fine-tuned to solve tasks related to that data modality. In relation to PD sleep-stage classification, we have in mind that one would train contrastive masked-token prediction models on all available LFP data (from PD patients and non-PD individuals), and then later fine-tune that model for the PD sleep-stage categorization task using the smaller volume of labeled data. We are currently pursuing this research strategy and hope to soon have results to report[35]. Furthermore, foundation models may be useful to derive frequency band boundaries used for sleep stage detection. After learning to extract predictive features from LFP data, the foundation model could be fine-tuned for classification using the relatively small amount of labeled data from PD patients. The recent successes of foundation models in language and[36] image identification[37] tasks suggest that this is a promising avenue of study. In the initial stage of foundation model development, we will use an unsupervised deep learning foundation model trained on the unprocessed, continuous time series LFP data to autonomously learn (i.e., in a self-supervised manner) a hierarchy of features without sleep epoch labels. These learned features will serve as a generalized representation of the data, and the model will then be fine-tuned using labeled data for specific sleep-stage classification[35].

One potential challenge in deploying these algorithms is the question of whether they perform well on new patients whose data were not included in the training set. In our experiments here, we found that all the models performed substantially worse when evaluated in a leave-one-out fashion, wherein nine patients' datasets were included in the training set and the model was tested on data from the remaining held-out patient dataset. We note that this analysis identified an overestimation (approximately 7% over actual predictive accuracy) in the findings reported by Christensen et al.[15]: In other words, the reported accuracy in the leave-one-out analysis was erroneously elevated

due to an analysis error. Our current algorithms do not perform as well on previously unseen patient datasets as on patients represented within the training set. This means that to deploy the algorithms in the adaptive DBS device, some labeled data may need to be obtained from each new patient to update the classifier model for that patient. Given the recent advances in foundation models[35,36], showing that qualitatively better generalization—including generalization to entirely new types of tasks—is achieved when models are trained on very large datasets, this limitation could be overcome by future work that uses larger volumes of LFP data. As discussed above, this could potentially make use of pre-training with unlabeled data, thereby reducing the quantity of labeled data that is required to achieve good generalization between individuals.

A critical limitation of our approach to model assessment for sleep stage classification is the limited capacity for assessing model performance in real-world neurophysiological conditions which includes the combined use of dopaminergic medication and therapeutic stimulation. Our prior work demonstrated that reliable sleep stage classification was not impeded by typical medication regimens in the absence of therapeutic DBS. Although in the present study we did not determine the impact of stimulation on our models' ability to infer sleep stages, it is reasonable to hypothesize that simulation will significantly affect local LFP signatures and thus classifier performance: recent studies have shown a dose-dependent response in overall beta LFP suppression with increasing DBS current amplitudes[37–39].

With consideration of real-life deployment of this large NN on theoretical IPGs, we assume that model training will be performed offline, and the resulting model will be embedded in the IPG for use. In the case of a deployed trained model, given that the inference time of our model is relatively quick (<1 s), the processing demands would largely depend on memory resources as the weights and architecture of our defined input data will require storage for execution and NN predictions. Another critical consideration for real-life deployment pertains to the performance metrics upon which algorithmic decisions are executed. We report accuracy as well as providing confusion matrices. However, it is important to consider the strength and confidence of the prediction, and relevance of metric when a model is used to change therapy. In future iterations of this work using a NN to effect changes in stimulation based on sleep stage classification, we aim to implement confidence thresholds that prioritize awake-type stimulation to mitigate risk of harm from false positives of non-awake states. Finally, an important question for the practical deployment of adaptive DBS devices is to identify the smallest and simplest model that can suffice for the adaptive DBS application, for the purposes of reducing computational and energetic cost. However, this was not the goal of the current study, in which we used externalized DBS leads, with stimulation settings adjusted by a computer with much greater processing power than those of commercially available implanted pulse generators. Thus, our current focus was on determining the efficacy of the overall approach, without substantial concern for hardware limitations. Once efficacy has been established, a further critical step will be to minimize the computation needed to obtain reasonable performance with the implanted hardware. Based on these considerations, the current study focused on identifying the best models. Important future work will investigate the performance-complexity tradeoff.

Altogether, our study points to the feasibility of adaptive DBS targeted at specific NREM sleep stages and suggests several paths forward to further optimize the categorization algorithms that will underlie such a treatment strategy, enabling new treatments that could reduce the sleep deficits suffered by PD patients.

## Data availability

The data that support the findings of this study are available from the corresponding author upon reasonable request.

## Code availability

The custom code for this work is available on GitHub at https://github.com/UH3-RestoreSleepPD/sleep_net.

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

## Acknowledgements

We are grateful for the participation of the patients in this study.

## Author contributions

K.C.: Methodology, Formal analysis, writing—review & editing; KS: writing—review & editing; E.C.: Methodology, writing—review & editing; A.A.: Conceptualization, Funding Acquisition, Data Curation, Investigation, Writing—review & editing; J.Z.: Conceptualization, Investigation, Writing—review & editing, Writing—Original draft, Formal Analysis, Methodology, Supervision; J.A.T.: Conceptualization, Funding acquisition, Investigation, Writing—review & editing, Writing—Original draft, Formal Analysis, Methodology, Supervision, Visualization, Project Administration.

## Competing interests

A.A. and J.A.T. receive research support from Medtronic, the remaining authors declare no competing interests.
