## [Peer Review File · Communications Engineering]

Reviewers' comments:

Reviewer #1 (Remarks to the Author):

This work showed an investigation to develop neural networks for classifying sleep stages in Parkinson's disease patients using signals available to the deep brain stimulation device. However, I have a number of further comments to improve the manuscript:

Introduction:

- Why are neural networks employed for categorizing NREM sleep? The rationale for this choice should be elucidated with reference to relevant literature, especially considering other machine learning techniques not only SVM and DT.
- Please maintain consistency in referring to both 'neural networks' and 'artificial neural networks,' as they essentially refer to the same concept.

Methods:

- Why is it necessary to normalize feature values, and what specific normalization technique is employed? Please provide a detailed explanation.
- What is the total duration of sleep utilized in this study? Please include an explanation of this in the Polysomnography Scoring section.
- The results of the SHapley Additive Explanations (SHAP) analysis are unclear regarding why NREM 2 and NREM 3 exhibit similar outcomes, despite being distinct types of sleep.

Results:

- It's better to show the sleep-stage classification process by indicating how the EEG signal is processed step-by-step, using the data from one subject as an example.

Discussion:

- For those data that are mis-classified, any analysis on the reasons?
- The performance of the proposed sleep-stage classification should be demonstrated in terms of accuracy, specificity, and sensitivity for different stages, ranging from awake to NREM 1, NREM 2, NREM 3, and REM. It would be benefit to present this information in a concise summary table.

Reviewer #2 (Remarks to the Author):

In the present manuscript, Carver et al. present an interesting analysis approach towards an improved sleep classification based on subthalamic local field potential data. They build on previous work, in which they already presented sleep stage classification based on the same dataset with a similar technique. However, in the present study, a more refined analysis of the non-REM-sleep-phases and an analysis aiming at applicability in chronically available recordings at a sampling rate of 250 Hz were added.

Combined subthalamic local field potential and polysomnography recordings with an expert scorer for

the sleep stages were used. For the models, NN were used that were trained on 75% of the epochs. Additionally, SHAP analysis was used to investigate the individual frequency bands' impact on the sleep stage classification.

As a result, the comparison between the different used models is mainly presented, with the main outcome that the larger model yields a higher accuracy in sleep stage prediction. Furthermore, the individual frequency bands are assessed with the result that especially low gamma and high beta contribute to the model.

The authors conclude that LFP based sleep staging using NN is feasible and could be used for adaptive algorithms.

Overall, this is an interesting study and a well-written manuscript using a sound computational approach for improved sleep staging in long high-resolution LFP-recordings with PSG and expert rating.

However, some more detailed information could be provided:

- Abstract:

o I found the abstract to be not very clear for representation of the study. structure could be improved, e.g. by putting more emphasis on results.

- Introduction:

o The introduction is nicely written, especially regarding the previous work and model discussion. However, if arguing towards using this in adaptive DBS, the potential benefit of sub-staging Non-REM sleep remains unclear. Please elaborate on this.

Also, if arguing towards the potential of the present method for aDBS, the currently available sleep-aware aDBS should be briefly discussed, e.g.:

♣ Gilron et al., 2021: Sleep-Aware Adaptive Deep Brain Stimulation Control: Chronic Use at Home With Dual Independent Linear Discriminate Detectors

- Results:

o Line 203-204 remain unclear – why a “but” in Line 204?

o Figure 1: It looks as if there have been two patients who have hardly slept at all. Is model performance different when 5 and 3 are excluded, albeit seeming to having very variable error rates?

o Could you please in the text give a more detailed description of how well the large model performed in the 4-stage- and 3-stage-differentiation? For example, what were the rates by sleep stage, especially for the distinction between N1 and N2/3?

o With current available chronic sensing-enabled devices, there may be a limitation to single frequency bands of interest – how well would individual bands such as low-gamma or high beta, or the two combined, perform in the model?

- Discussion

o Furthermore, these recordings were performed in a relative OFF-medication state and without stimulation. This is, of course, not the expected therapeutic state for assessment of sleep architecture during aDBS. Please comment on the expected impact of a full medication regime and DBS therapy a) to the signal and b) to the sleep architecture, given the current observation the even in the controlled settings, intraindividual differences lead to a decreased model performance.

Reviewer #3 (Remarks to the Author):

Summary

The authors present a compelling case for the feasibility of using small neural net classifiers for automatic sleep staging in implantable neurostimulation systems. Broad clinical consensus exists to the relevance of sleep/wake adjustments to deep brain stimulation as a therapy refinement. The Tiny ML research space is also currently under-explored in the context of neuroscience, making this paper both timely and of great interest to the clinical neuroscience and neural engineering communities.

The authors uncover a very exciting and potentially counterintuitive result, suggesting that larger neural networks are capable of compensating for the performance loss incurred due to low sampling rates. The conclusions are supported by a model interpretation effort via Shapley value attribution, which I found helpful in building intuition about the presented classifiers.

The manuscript is well-written, and is an appropriate length for the research questions covered. The methods used are sound, and the supporting figures are clear to understand.

At present the paper takes sampling rate constraints in implantable pulse generators as the main motivation for the following model exploration, however this limitation of IPGs is not universal: higher sampling rate IPGs exist. IPGs however, are without exception short on computational capacity and are designed in a fine balance of computational complexity and battery life. As stated, the authors uncovered a novel relation between algorithm complexity and sampling rate for designing neural network classifiers: the exploration of this trade-off is in itself valuable, and merits publication in my opinion. I recommend that the introduction and discussion be reframed to emphasise the trade-off as the key result, as opposed to focusing solely on the feasibility of low sampling rate operation. This would increase the impact of the paper by informing the design of next generation IPGs. This reframing can be achieved with relatively minor effort, and could be supported by including an additional 'medium' sampling rate model.

Comments

Line 59

"inference of sleep stage from deep brain signals (10)"

This should be more specific, sleep staging in EEG, ECoG has been established longer than in deep brain targets. Reference 14, Thompson et al. 2018, that the authors already use could be introduced earlier to give further support to this claim (currently first appears on line 75).

Lines 69, 232, 255, 281

"current commercially-available DBS platforms record LFP signals at only 250 Hz"

The authors currently rely on reference 12 for this claim (Jimenez-Shahed 2021), which seems to solely describe the Medtronic Percept PC. The Medtronic Summit RC+S could be configured for 1000 Hz sampling (Gilron et al. 2021, doi: 10.1038/s41587-021-00897-5), the Picostim-DyNeuMo is reported at 625 Hz (Toth et al. 2020, doi: 10.1109/SMC42975.2020.9283187), while the Newronika AlphaDBS is reportedly capable of 512 Hz though band-limited to 40 Hz, presumably for improved SNR in the beta-band (Arlotti et al. 2021, doi: 10.3389/fnins.2021.763235).

The feasibility of low-sampling rate classification is relevant irrespective of device capabilities for several other reasons (e.g. power savings to offset the computational overhead of the algorithm), and these should be emphasised instead of focusing on the rather restrictive set of present-day commercial devices. It is also uncertain whether commercial devices have the computational resources to run the proposed algorithms (see further comments on Discussion). The paper can make a valuable contribution to the field by analysing the algorithm complexity vs sampling rate trade off and focus on making a recommendation to guide the hardware specifications of upcoming IPGs.

Line 70

"classification of sleep stages from intracranial recordings has been explored in epilepsy"

(1) To the previous point on validation of STN staging – does intracranial refer to deep brain or cortical areas here?

(2) The authors should also discuss to what extent sleep stage-level granularity of stimulation control is likely to carry benefits over simpler approaches, such as circadian scheduling for approximating sleep/wake. This simpler approach is used in LivaNova's SenTiva IPG, or experimentally in the Picostim-DyNeuMo (Zamora et al. 2021, doi: 10.3389/fnins.2021.734265).

Line 80

"prediction accuracy"

The authors helpfully report confusion matrices along with accuracy as a summary metric of their model. A discussion point on the relative importance of different classification errors would strengthen the paper. In some contexts e.g. responsive neurostimulation in epilepsy false positive seizure detections are more acceptable than false negatives. It is not immediately clear what the right balance is in sleep staging.

Line 142

"Sleep stages were determined by analysis of 30-s epochs of the PSG, by a sleep neurologist"

The authors do point out the limitation of performing sleep staging by only a single specialist later in the text (line 289 onwards), and rightfully conclude that this limitation should not detract from the results. It might be helpful for future readers to have a forward reference to this discussion.

Line 172

"A limited hyperparameter search was performed, which included the addition of another hidden layer, and reducing or increasing the width of the hidden layers by a factor of 2: These changes did not substantially affect model accuracy."

Why not report the smaller model if accuracy is not substantially different? Model complexity will ultimately have to be balanced with performance to fit the algorithm into the constraints of an IPG, this warrants discussion.

Line 226

Missing punctuation

Lines 255 and 281

"the high gamma frequency band is reduced in currently available (IPGs)"

In addition to the question of sampling rate (previously discussed), at least a brief mention of noise-floor in IPGs should be included. Neural signals follow $1/f$ power, while the IPG recording front-end will have its own noise floor, typically poorer than an externalised recording system. Is there an expectation based on IPG noise profiles on

Line 301

"the dearth of non-REM 3 epochs"

How could this be addressed? Is the only solution to collect more patient data? Could synthetic data approaches be of use here? Foundation models (as discussed next)?

Line 312

"Future studies aiming to advance this goal could benefit from foundation model approaches"

A definition for foundation models should be included. In the current form it is slightly unclear that "training models using unsupervised predictive loss functions on larger reams of potentially unlabeled LFP data from both PD patients and other population" is an implicit definition.

Line 318

"After learning to extract predictive features from LFP data, the foundation model could be fine-tuned for classification"

I suspect the step of unsupervised learning to extract predictive features might be new to some readers interested in this paper. It could be worth clarifying why this is possible, or providing an additional reference.

Line 322

"One potential challenge in deploying these algorithms is the question of whether they perform well on new patients whose data were not included in the training set. In our experiments here, we found that all the models performed substantially worse when evaluated in a leave-one-out fashion, wherein nine patients' datasets were included in the training set and the model was tested on data from the remaining held-out patient dataset."

(1) This seems to suggest 9-10 patients are not sufficient to capture the full heterogeneity/variance of the population. Could the authors estimate how much data would be necessary to achieve this? As a practical matter, there might be a regulatory difference in deploying models with patient specific training, that could make it preferable to use a single model trained on a larger sample.

(2) As a second point, this indicates results might be sensitive to the exact training data. The paper describes using 25% data held out for validation. Is there a reason cross-validation is not practical here? If it is feasible to perform, it could strengthen the results of the paper by mitigating the data-dependency.

Further comments for Discussion

(1)

A cursory search for "STN sleep staging" reveals another paper which reports very similar headline accuracy (91%) using a support vector machine (Chen et al. 2019, doi: 10.1109/TNSRE.2018.2890272). As a significantly different approach, it could be a valuable point of comparison and contrast for the merits of this paper on a neural net approach.

(2)

The Discussion should return to the feasibility of deploying the proposed classifier in a real-world IPG. In lack of a specific IPG to target, performing quantization and other explicit steps in complexity reduction are not necessary as no target model size is available; qualitative comments on whether the classifier is e.g. memory / processor heavy may suffice. Should the authors be interested, some example memory and clock cycle estimates for a small neural net are made in Kavoosi et al. 2022, doi: 10.1109/EMBC48229.2022.9871793

(3)

The authors should consider briefly reporting on 'intermediate' sampling rate models to flesh-out the complexity trade-off, e.g. would a 400 Hz model have all the benefits of including the high-gamma band, while requiring fewer units / labels? Or a 700 Hz model that includes the HFO band?

We appreciate the invaluable feedback provided by the reviewers. Their insights and suggestions have greatly contributed to enhancing the quality of our work. We have taken all comments into careful consideration and have made the corresponding revisions to the manuscript. Below, we outline how each comment has been addressed, and we hope that these revisions adequately address the reviewers' concerns and queries for clarification.

Reviewers' comments:

Reviewer #1 (Remarks to the Author):

1. Introduction:

Why are neural networks employed for categorizing NREM sleep? The rationale for this choice should be elucidated with reference to relevant literature, especially considering other machine learning techniques not only SVM and DT.

This is an excellent point of clarification as there are several machine learning strategies that have been and could be used for sleep stage classification. Part of the rationale for our use of a neural network was based on the poor performance of other methods, including SVM (Thompson et al., 2018). Classifying NREM states is a challenging problem as the ground truth labeling depends on manual scoring by humans and is not computationally quantifiable. Our study is further complicated by two factors: 1) Parkinson's disease subjects have dysregulated sleep stage representation resulting in difficult expert assessment of their sleep behavior and 2) they exhibit a decrement and, in some cases, a deficit in NREM stage 3 and REM. These factors, in combination, pose a significant challenge for classification. We note on lines XX, Page 2, that our use of SVM was insufficient and allude to some of the challenges listed in this response. We opted for a neural network approach for the following reasons: 1) they are better at handling high-dimensional and complex data, of which our local field potential data sets are composed, and 2) they have greater end-to-end learning and flexibility. We posit that an NN approach will be better at learning the necessary transformations and mappings in an automated fashion, particularly in the context of the data challenges described. We have added the following to lines 102-104,

"Following our prior experience with other machine learning approaches, we opted to use an NN approach for the advantages associated with better handling of complex data and greater end-to-end learning and flexibility."

2. Please maintain consistency in referring to both 'neural networks' and 'artificial neural networks,' as they essentially refer to the same concept.

We appreciate this point of clarification and consistency. We have opted to remove all references to ANN and have used NN throughout the manuscript to refer to neural networks.

3. Methods:

Why is it necessary to normalize feature values, and what specific normalization technique is employed? Please provide a detailed explanation.

LFP signals have different amplitudes for different subjects. Without normalization, the magnitudes of the activations within the neural network are subject-specific which confounds the network's ability to

detect sleep-stage-specific signals. Our aim is to make algorithms that return sleep-stage-specific outputs (e.g., categorize sleep stages) in a manner that is invariant to the subject from which the data were obtained: i.e., we want the correct sleep stage label to be returned by the algorithm, regardless of the subject from whom the data were obtained. For this reason, our very earliest efforts to train neural networks on the “raw” LFP signals without normalization (as part of our published 2019 Christensen et al. study) obtained relatively poor sleep stage categorization accuracy. When we incorporated the LFP signal normalization into the data preprocessing, it removed this extraneous factor of variation in the data, and much better performance was obtained. For this process we used a standard z-scoring procedure (mean of 0 and standard deviation of 1) in the data preprocessing pipeline.

It is worth noting that normalizing input signal amplitudes is a standard approach in machine learning, (e.g., One typically normalizes the pixel values in images when inputting them into object recognition networks, using either batch normalization, or layer normalization), for the reasons discussed above. See line 197-201 of Methods.

4. What is the total duration of sleep utilized in this study? Please include an explanation of this in the Polysomnography Scoring section.

This is an excellent point of clarification. Based on the polysomnography scoring we used the following to calculate total sleep: we took the sum of all sleep stages excluding periods of wakefulness, from the time the individual first falls asleep to the time of final awakening. We have added the following text to the manuscript on lines 172-175.

“Expert derived sleep stages, based on polysomnography evaluation, were used to calculate total duration of sleep over the period of one night. To estimate total duration of sleep, scored epochs were summed excluding periods of wakefulness, from the time the individual first falls asleep to the time of final awakening; for details, please see Figure 1 of Christensen et al., 2019.”

5. The results of the SHapley Additive Explanations (SHAP) analysis are unclear regarding why NREM 2 and NREM 3 exhibit similar outcomes, despite being distinct types of sleep.

This is an excellent clarification question. We agree with the reviewer that NREM 2 and NREM 3 represent distinct types of sleep. However, due to the critical dearth of NREM 3 epochs observed in our PD subject cohort, we decided to combine NREM 2 and NREM 3 states into one label, NREM2/3, in our model. Thus, there is no separate SHAP analysis performed for NREM 2 and NREM 3; there is only the analysis performed on the combined 2/3 label thus only one outcome representing the two states.

6. Results:

It's better to show the sleep-stage classification process by indicating how the EEG signal is processed step-by-step, using the data from one subject as an example.

We thank the reviewer for this clarification question. If the reviewer is referring to the sleep stage classification from the electroencephalogram (EEG) included in the polysomnography assessment, then no non-standard steps were taken to process those data. Per AASM procedure for sleep scoring, the sleep expert reviewer would examine the EEG signals along with electrooculogram (EOG), electromyography (EMG), and video data to assess sleep stage classification and video data to assess sleep stage classification. Detailed information about sleep stage classification from EEG data is detailed in Thompson et al 2018. If the reviewer is referring to the LFP processing steps that we used to prepare

data for sleep stage classification with our neural network, then in addition to the methods description we have included a schematic of the processing steps – See revised Figure 1C.

7. Discussion:

For those state/data that are mis-classified, any analysis on the reasons?

There are several possible causes for misclassification of any given epoch. One issue is that the ground truth labels in this study were obtained from a single sleep expert, and sleep staging is known to have some ambiguity: when different human experts label epochs using the same data from the same night's sleep, they agree on about 83% of the epochs [R.S. Rosenberg, S. Van Hout 2013 – J Clinical Sleep Medicine]. This provides an estimate on how well-defined the sleep labels are and sets a ceiling on how well our models can realistically perform. Inter-rater reliability is highest for REM sleep stages and lowest for N1 and N3 stages. This means that these sleep stages are more challenging to identify, and our modeling results echo that variation in difficulty level: from newly added Table 1 (4 Classes Large N) and the confusion matrix in Fig. 4C.

8. The performance of the proposed sleep-stage classification should be demonstrated in terms of accuracy, specificity, and sensitivity for different stages, ranging from awake to NREM 1, NREM 2, NREM 3, and REM. It would be benefit to present this information in a concise summary table.

We thank the reviewer for this suggestion. We have created a new Table 1 to represent these summary metrics for all models and sleep stages.

Reviewer #2 (Remarks to the Author):

1. Abstract:

I found the abstract to be not very clear for representation of the study. structure could be improved, e.g. by putting more emphasis on results.

We thank the reviewer for this suggestion and have revised the Abstract for clarity – See revised Abstract. The journal limits the abstract to 200 words.

“Sleep dysfunction affects over 90% of Parkinson’s Disease (PD) patients. Recently, subthalamic nucleus (STN) deep brain stimulation (DBS) has shown promise for alleviating sleep dysfunction. We sought to improve DBS for sleep dysfunction through sleep-state-specific neuromodulation. We previously showed that a small neural network (NN) could classify sleep stages from STN local field potential (LFP) recordings in PD patients. This previous NN could categorize three states: awake, rapid-eye-movement (REM), and non-rapid eye-movement (NREM). However, it was unable to divide NREM into its different sub-categories, NREM 1, 2, and 3. To address this limitation, in our current study we demonstrate that a larger NN architecture can sub-categorize two NREM stages (NREM1 and a combined NREM2&3 label) for the first time and with reasonable accuracy, up to 88% for NREM1 and 92% for NREM2&3. Using Shapley attribution analysis, we determined the contributions across different frequency bands of STN LFP signals to sleep stage categorization and found that middle-to-high-frequency bands (low gamma, high beta) are more important to model decision than the lowest and highest frequency bands. These results suggest that our proposed NN-based adaptive DBS treatment can indeed be implemented in commercially available devices with lower LFP sampling frequencies.”

2. Introduction:

The introduction is nicely written, especially regarding the previous work and model discussion. However, if arguing towards using this in adaptive DBS, the potential benefit of sub-staging Non-REM sleep remains unclear. Please elaborate on this.

This is an excellent point of clarification regarding the potential benefits of sub-staging NREM for adaptive DBS. In the sleep literature focused on the health benefits related to enhanced slow-wave (NREM) sleep, there is considerable ongoing effort to selectively target those sleep stages for various forms of stimulation. In Parkinson's Disease-associated sleep dysfunction, specific sleep stages such as NREM 3 exhibit reduced expression. Slow-wave sleep is identified as the most restorative sleep stage and is associated with sleep maintenance and quality [Dijk et al., 2019]; targeting slow wave sleep using transcranial electrical stimulation in healthy populations showed an increase in time spent in slow-wave sleep and an increase in sleep-mediated declarative memory [Grimaldi et al., 2020]. We hypothesize that the ability to identify sleep stages, particularly sub-NREM stages, will provide necessary for delivering optimal stimulation. We have added the following to the end of the Introduction (Lines 124-128):

"On-going efforts to selectively target specific sleep states for external (Grimaldi et al., 2020; Dijk et al., 2009) and internal (Smyth et al., 2023) stimulation to improve overall sleep quality will benefit from this work. With regards to PD, stage-targeted stimulation may improve NREM duration and expression, which is implicated in restorative rest and overall sleep maintenance and quality [Dijk et al., 2009; Smyth et al., 2023]."

3. Also, if arguing towards the potential of the present method for aDBS, the currently available sleep-aware aDBS should be briefly discussed, e.g.:
Gilron et al., 2021: Sleep-Aware Adaptive Deep Brain Stimulation Control: Chronic Use at Home With Dual Independent Linear Discriminate Detectors

We thank the reviewer for bringing this reference to our attention. We have incorporated a brief discussion of the main findings as they relate to our work in the Introduction. We have added the following to lines 115-117:

"Prior work in PD subjects treated with DBS have used an embedded linear classifier to determine asleep and awake states demonstrating the feasibility of using in-home LFP assessment for aDBS (Gilron et al., 2021)"

4. Results:

Line 203-204 remain unclear – why a “but” in Line 204? ----

We appreciate the reviewer noting this unclear section.

We have revised that section.

5. Figure 1: It looks as if there have been two patients who have hardly slept at all. Is model performance different when 5 and 3 are excluded, albeit seeming to having very variable error rates?

Participants 5 and 3 did indeed have fewer sleep epochs than the other participants. At the same time, the classifier's error rates on those participants were not significantly different than on the other participants (e.g., see Fig. 3BD, which shows the error rate for each subject).

6. Could you please in the text give a more detailed description of how well the large model performed in the 4-stage- and 3-stage-differentiation? For example, what were the rates by sleep stage, especially for the distinction between N1 and N2/3?

We thank the reviewer for this request for clarification in the presentation of the results. We include a new Table to provide an accessible summary of the model results by sleep stage. Please see newly added Table 1.

7. With current available chronic sensing-enabled devices, there may be a limitation to single frequency bands of interest – how well would individual bands such as low-gamma or high beta, or the two combined, perform in the model?

This is an excellent suggestion. In response to this query, we have trained and tested 3 new models with the following input data: 1) only high-beta power data, 2) only low-gamma power data and 3) combined low-gamma and high-beta power data. We find that overall, sub-bands perform poorly when compared to the full complement of frequency band information – all assessed metrics dropped by 15-20%, compared to the full data set. We have included a new Figure – Figure 6A that represents the results of this analysis. In addition, we have added text in the Results describing these analyses and observations. See Lines 302-308.

8. Discussion

Furthermore, these recordings were performed in a relative OFF-medication state and without stimulation. This is, of course, not the expected therapeutic state for assessment of sleep architecture during aDBS. Please comment on the expected impact of a full medication regime and DBS therapy a) to the signal and b) to the sleep architecture, given the current observation the even in the controlled settings, intraindividual differences lead to a decreased model performance.

This is a great point of clarification for our consideration of real-world implementation of our proposed strategy. We have expanded our limitation section to include a discussion of the challenges to our model based on the potential impact of more variable real-world conditions including PD medication and therapeutic stimulation. We have added the following to the discussion section See lines 391-398:

“A critical limitation of our approach to model assessment for sleep stage classification is the limited capacity for assessing model performance in real-world neurophysiological conditions which would include dopaminergic medication and therapeutic stimulation. Our prior work demonstrated that reliable sleep stage classification was not impeded by typical medication regimens, in the absence of therapeutic DBS. Although, in this study we did not determine the predictive impact of stimulation, it is reasonable to hypothesize that that stimulation will invariably affect local LFP signatures, as recent studies have shown a dose dependent response in overall beta LFP suppression with increasing DBS amplitudes (Feldmann 2021 and 2022).”

Reviewer #3 (Remarks to the Author):

1. Line 59

"inference of sleep stage from deep brain signals (10)"

This should be more specific, sleep staging in EEG, ECoG has been established longer than in deep brain targets. Reference 14, Thompson et al. 2018, that the authors already use could be introduced earlier to give further support to this claim (currently first appears on line 75).

This is an excellent suggested improvement and we have revised the text to reflect a more nuanced discussion of sleep stage inference from brain signals with various modalities. In addition, we have referenced our prior work at this same point in the manuscript, which is earlier than in the original submission. We have expanded the text on line 62-63 to the following new text:

“Direct sensing of local field potential (LFP) data from the DBS electrodes has recently become commercially available, enabling the inference of sleep stage from deep brain signals (10), which will build from established work in ECoG, EEG and acute DBS recordings (Thompson et al., 2018 and other references).”

2. Lines 69, 232, 255, 281

"current commercially-available DBS platforms record LFP signals at only 250 Hz"

The authors currently rely on reference 12 for this claim (Jimenez-Shahed 2021), which seems to solely describe the Medtronic Percept PC. The Medtronic Summit RC+S could be configured for 1000 Hz sampling (Gilron et al. 2021, doi: 10.1038/s41587-021-00897-5), the Picostim-DyNeuMo is reported at 625 Hz (Toth et al. 2020, doi: 10.1109/SMC42975.2020.9283187), while the Newronika AlphaDBS is reportedly capable of 512 Hz though band-limited to 40 Hz, presumably for improved SNR in the beta-band (Arlotti et al. 2021, doi: 10.3389/fnins.2021.763235).

The feasibility of low-sampling rate classification is relevant irrespective of device capabilities for several other reasons (e.g. power savings to offset the computational overhead of the algorithm), and these should be emphasised instead of focusing on the rather restrictive set of present-day commercial devices. It is also uncertain whether commercial devices have the computational resources to run the proposed algorithms (see further comments on Discussion). The paper can make a valuable contribution to the field by analysing the algorithm complexity vs sampling rate trade off and focus on making a recommendation to guide the hardware specifications of upcoming IPGs.

We agree with the reviewer and thank them for this excellent suggestion! We have modified the way this work is presented in the text, to instead ask the question of whether the higher frequency bands are really needed, and whether a simpler model (relying only on the lower frequencies) might suffice.

Lines 73-80:

“Our secondary aim was to understand whether the sleep stage categorization can be accomplished using LFP data gathered at lower sampling rates: while the dataset studied in our previous work was collected at 1024 Hz, current commercially available DBS platforms record LFP signals at ranges of 250 to

1000 Hz (12). For this study, we decided to consider the lowest sampling rate — 250Hz — when creating the downsampled model for two reasons: firstly, to determine if our approach would work on the broadest range of DBS devices currently available, where 250Hz is the strictest sampling floor; and secondly, to determine how much classification performance benefits from the use of higher sampling rates.”

Lines 258-265:

3.3 Impact of downsampling on four sleep stage label prediction

The models described above were trained and tested using LFP data that were recorded at 1024 Hz, which included the high gamma (90-200 Hz) band and the high frequency oscillations (HFO, 200-350 Hz). However, currently available FDA approved DBS implanted programmable generators (IPGs) that permit recording from brain leads are limited to 250 Hz sampling resolution. Given the Nyquist limit, this means that IPGs capable of recording can resolve LFP signals up to 125 Hz. Furthermore, while other DBS devices with higher sampling resolutions exist, we again focus on the most limiting resolution case to understand how much sleep stage classification performance is degraded in cases where sampling resolution is limited.

Lines 284-293 (SHAP analysis):

For the Awake sleep stage, low gamma contributed most to the model prediction (Figure 5A), and for non-REM 1, combined non-REM 2 & 3, and REM, high beta was the highest contributor to the model prediction (Figure 5 B, C, and D). Figure 5E depicts the relative contributions of the LFP bands, highlighting that high beta, low gamma and high gamma contributed most to predictions on average. Notably, the high gamma frequency band, which was only included in the top three features of the Awake label, is reduced in currently available FDA-approved implantable pulse generators, which capture high gamma signals up to 125 Hz. The relative unimportance of the high gamma band and other low-frequency bands such as delta and alpha suggest that a model using an even greater constraint of feature bands may be possible while retaining accuracy. Here, we classify high gamma signals up to 200 Hz (and HFOs up to 350 Hz) in our full resolution data.

Discussion Lines 321-331:

In addition to the question of non-REM sub-classification - a challenge posed by electrophysiological signal ambiguity, model complexity, and limited expression of non-REM3 - we also addressed the practical question of whether LFP-based sleep stage categorization could succeed using LFP signals collected with existing commercially available hardware and sampling constraints (i.e., 250 Hz limit). Compared with higher-resolution LFP data (1024 Hz) collected with a clinical neurophysiological recording system, we saw a moderate loss of classification performance when the lower-resolution LFP data were used for the classification. While our results suggest that adaptive DBS treatments could potentially work with lower temporal resolution LFP recordings, they also indicate that these treatments could benefit from increasing the temporal resolution of the devices' LFP sampling: Specifically, classification performance was reasonable even with downsampled LFP data but was substantially higher when a larger range of input frequencies was used.

3. Line 70

"classification of sleep stages from intracranial recordings has been explored in epilepsy"

(1) To the previous point on validation of STN staging – does intracranial refer to deep brain or cortical areas here?

This is also an excellent point of clarification; we appreciate this highlight from the reviewer. We have included expanded text to Line 81-82, which now reads as follows:

'In addition to basal ganglia recordings in PD, automatic classification of sleep stages from intracranial recordings, both cortical and hippocampal, have been explored in epilepsy.'

4. (2) The authors should also discuss to what extent sleep stage-level granularity of stimulation control is likely to carry benefits over simpler approaches, such as circadian scheduling for approximating sleep/wake. This simpler approach is used in LivaNova's SenTiva IPG, or experimentally in the Picostim-DyNeuMo (Zamora et al. 2021, doi: 10.3389/fnins.2021.734265).

We appreciate the reviewer bringing this point to our attention. Several recent articles have observed clear multi-scale biorhythms in STN recorded from PD patients treated via DBS which could be used for stimulation control as indicated by the reviewer. We have amended the text to incorporate this excellent suggestion as an adjunctive to our neural network approach. We have added the following to Lines 86-90:

"In addition to targeted sleep stage classification, recent efforts in PD and epilepsy have observed multi-scale biorhythms that could be used to inform adaptive stimulation control schemes (Zamora et al., 2021). Although our current work is predicated on single-stage resolution for intervention, one or more diurnal neural biorhythms could be explored as an adjunctive strategy."

5. Line 80

"prediction accuracy"

The authors helpfully report confusion matrices along with accuracy as a summary metric of their model. A discussion point on the relative importance of different classification errors would strengthen the paper. In some contexts e.g. responsive neurostimulation in epilepsy false positive seizure detections are more acceptable than false negatives. It is not immediately clear what the right balance is in sleep staging.

This is an excellent point of clarification suggested by the reviewer. In the context of an algorithm or model conferring clinical decisions, particularly decisions that deviate from therapeutic thresholds, it is important to consider the relevance and confidence of the prediction metric. We have amended the

Discussion section which now includes a brief treatment of this important consideration. We have added the following on Lines 405-409:

“For model performance, we reported accuracy as well as provided confusion matrices. However, it is important to consider the strength and confidence of the prediction, and relevance of metric when a model is used to change therapy. In future iterations of this work, in which a neural network is used to effect changes in stimulation based on sleep stage classification, we aim to implement confidence thresholds that prioritize awake-type stimulation to mitigate risk of harm from false positives of non-awake states.”

6. Line 142

"Sleep stages were determined by analysis of 30-s epochs of the PSG, by a sleep neurologist"
The authors do point out the limitation of performing sleep staging by only a single specialist later in the text (line 289 onwards), and rightfully conclude that this limitation should not detract from the results. It might be helpful for future readers to have a forward reference to this discussion.

We thank the reviewer for this clarification. We have included added the following reference to support this discussion [West et al., 2023 ‘Evaluation of consensus sleep stage scoring of dysregulated sleep in Parkinson’s disease’] Lines: 173-175;

7. Line 172

"A limited hyperparameter search was performed, which included the addition of another hidden layer, and reducing or increasing the width of the hidden layers by a factor of 2: These changes did not substantially affect model accuracy."

Why not report the smaller model if accuracy is not substantially different? Model complexity will ultimately have to be balanced with performance to fit the algorithm into the constraints of an IPG, this warrants discussion.

We agree with the reviewer that an important question is to identify the smallest and simplest model that can suffice for the adaptive DBS application in response, we have added the following paragraph to our Discussion section See Lines 409-419.

“An important question for the practical deployment of adaptive DBS devices is to identify the smallest and simplest model that can suffice for the adaptive DBS application. At the same time, that was not the goal of this study: our current clinical experiments use externalized leads, where the stimulation settings are adjusted by a computer much larger than that in the implanted pulse generator. Thus, our current focus was on determining the efficacy of the overall approach, without substantial concern for hardware

limitations. Once efficacy has been established, a further critical step would be to minimize the computation needed to obtain reasonable performance with the implanted hardware. Based on these considerations, the current study focused on identifying the best models: important future work will investigate the performance complexity tradeoff."

8. Line 226
Missing punctuation –

We thank the reviewer for bringing this missing punctuation to our attention. We have added the necessary period and sentence separation.

9. Lines 255 and 281
"the high gamma frequency band is reduced in currently available (IPGs)"

In addition to the question of sampling rate (previously discussed), at least a brief mention of noise-floor in IPGs should be included. Neural signals follow 1/f power, while the IPG recording front-end will have its own noise floor, typically poorer than an externalised recording system. Is there an expectation based on IPG noise profiles on performance

We agree that this is an important limitation to note. We have added the following text to the Discussion on Lines 409-419. See text in response to item 7 above.

10. Line 301
"the dearth of non-REM 3 epochs"

How could this be addressed? Is the only solution to collect more patient data? Could synthetic data approaches be of use here? Foundation models (as discussed next)?

The shortage of non-REM 3 epochs in PD patients poses a serious challenge for the supervised learning approach we used in this paper, wherein we trained only on PD patient LFP data and associated sleep stage labels. In the foundation model approach – which we are actively pursuing – one would instead learn a generative model in a self-supervised manner using all the available LFP data (including data from PD patients and non-Parkinsonian individuals). This model then learns the inherent structure of LFP signals and can then be fine-tuned later for the categorization task on the size-limited dataset of labeled PD-patient data. Using this approach, the task of identifying non-REM 3 in PD patients would benefit from the availability of LFP signals from non-REM 3 in non-Parkinsonian patients (even if it is not labeled as such – recall that the larger dataset need not be labeled for use in the self-supervised training phase).

11. Line 312

"Future studies aiming to advance this goal could benefit from foundation model approaches"

A definition for foundation models should be included. In the current form it is slightly unclear that "training models using unsupervised predictive loss functions on larger reams of potentially unlabeled LFP data from both PD patients and other population" is an implicit definition.

We thank the reviewer for pointing this out and have added the following definition (see lines 359-365)

"We define a foundation model as a model that is first trained in a self-supervised manner using large quantities of unlabeled data, then is fine-tuned to solve tasks related to that data modality. In relation to PD sleep stage classification, we have in mind that one would train contrastive masked-token prediction models on all available LFP data (from PD patients and non-Parkinsonian individuals), and then later fine-tune that model for the PD sleep stage categorization task using the smaller volume of labeled data. We are currently pursuing this research strategy and hope to soon have results to report."

12. Line 318

"After learning to extract predictive features from LFP data, the foundation model could be fine-tuned for classification"

I suspect the step of unsupervised learning to extract predictive features might be new to some readers interested in this paper. It could be worth clarifying why this is possible, or providing an additional reference.

We agree with the reviewer's point that further clarification of our use of a foundation model for sleep stage classification would be helpful for readers. We have included the following text in the Discussion, see Lines 369-374.

"In the initial stage of model development, we will use an unsupervised deep learning foundation model trained on the unprocessed, continuous time series LFP data to autonomously learn (i.e., self-supervised) a hierarchy of features without sleep epoch labels. These learned features will serve as a generalized representation of the data, and the model will then be fine-tuned using labeled data for specific sleep stage classification (Bommasani et al., 2021)."

13. Line 322

"One potential challenge in deploying these algorithms is the question of whether they perform well on new patients whose data were not included in the training set. In our experiments here, we found that all the models performed substantially worse when evaluated in a leave-one-out fashion, wherein nine patients' datasets were included in the training set and the model was

tested on data from the remaining held-out patient dataset."

(1) This seems to suggest 9-10 patients are not sufficient to capture the full heterogeneity/variance of the population. Could the authors estimate how much data would be necessary to achieve this? As a practical matter, there might be a regulatory difference in deploying models with patient specific training, that could make it preferable to use a single model trained on a larger sample.

This is an excellent point of consideration for generalizability and feasibility in deployment. Based on our prior work in Christensen et al., 2019, using a leave-one-out approach we observed acceptable predictive performance across most subjects. Given the complexity of real-world conditions that our experimental design does not account for, it is likely that we have not explored the full parameters space of variability in sleep stage architecture, nor STN LFP variability exhibited in our PD cohort. In current, unpublished work in which we have acquired additional subjects with more than one night of recorded data, we see evidence for consistency in LFP signals and sleep structure for individual subjects and hypothesize that with more nights from individual subjects the model could be sufficiently improved.

14. (2) As a second point, this indicates results might be sensitive to the exact training data. The paper describes using 25% data held out for validation. Is there a reason cross-validation is not practical here? If it is feasible to perform, it could strengthen the results of the paper by mitigating the data-dependency.

We agree with the reviewer that there are advantages to using a k-fold cross validation approach, which we used in our earlier manuscript for the smaller models. However, we chose not to use this approach for the larger models as it was not practical for iterative training and testing of the models given the significant increase in the time added for the k-fold validation.

15. A cursory search for "STN sleep staging" reveals another paper which reports very similar headline accuracy (91%) using a support vector machine (Chen et al. 2019, doi: 10.1109/TNSRE.2018.2890272). As a significantly different approach, it could be a valuable point of comparison and contrast for the merits of this paper on a neural net approach.

This is an excellent point and opportunity to compare an alternate methodology. Both our approach and Chen et al are built using mathematically proven universal function approximators [1,2]. In fact, it is proven that a NN with a single hidden layer of infinite width is equivalent to an SVM and under easily met conditions a finite width NN is nearly equivalent to and approximates well an SVM [1]. NN have a few small practical differences that we think makes them a superior option for embedded devices. They are straightforward to implement in hardware, typically requiring only matrix multiplication operations as opposed to exponent operations necessary for the kernel used in Chen et al (radial basis filter, RBF), or an often even more complex polynomial kernel. NN often performs well under data-scarce training conditions, or environments where even small accuracy improvements are important (such as diagnostics or medical devices). However, SVM has a distinct advantage of being less sensitive to parameter initializations where NN may struggle to escape local minima in the loss space, though there

are training techniques to mitigate this. Finally, SVM models are often considered more “explainable” and less “black box” than NN in terms of understanding and evaluating how the model arrives at its classification.

[1] <https://openreview.net/forum?id=npUxA--nyX>

[2] <https://link.springer.com/article/10.1023/A:1022936519097>

16. The Discussion should return to the feasibility of deploying the proposed classifier in a real-world IPG. In lack of a specific IPG to target, performing quantization and other explicit steps in complexity reduction are not necessary as no target model size is available; qualitative comments on whether the classifier is e.g. memory / processor heavy may suffice. Should the authors be interested, some example memory and clock cycle estimates for a small neural net are made in Kavooosi et al. 2022, doi: 10.1109/EMBC48229.2022.9871793

This is an excellent suggestion for consideration. We have added text to the Discussion provided our qualitative assessment of the performance demands of our classifier in a theoretical real-world IPG. Please see lines 399-403. The following has been added to the manuscript.

“With consideration of real-life deployment of this large NN on theoretical IPGs, we assume that model training will be performed offline, and the model will be embedded in the IPG for use. In the case of a deployed trained model, given that the inference time of our model is relatively quick (< 1 second), the processing demands would largely depend on memory resources as the weights and architecture of our defined input data will require storage for execution and NN predictions.”

17. The authors should consider briefly reporting on 'intermediate' sampling rate models to flesh-out the complexity trade-off, e.g. would a 400 Hz model have all the benefits of including the high-gamma band, while requiring fewer units / labels? Or a 700 Hz model that includes the HFO band?

This is an excellent suggestion. In response to this query, we have trained and tested 3 new models (in addition to the full spectrum model (1024Hz and down sampled model at 250Hz) with the following input data: 1) 125Hz, 2) 400Hz and 3) 700Hz. We find that overall that there is minimal reduction in performance (1-2%) across metrics when higher frequencies are removed from the data set. We have included a new Figure – Figure 6B that represents the results of this analysis. In addition, we have added text in the Results describing these analyses and observations. See Lines 273-275.

REVIEWERS' COMMENTS:

Reviewer #1 (Remarks to the Author):

The authors well revised the manuscript taking into account all review's comments

Reviewer #2 (Remarks to the Author):

The authors have thoroughly and successfully dealt with my comments. They have added interesting additional analysis and comprehensive improvements to the figures. I think the overall quality of the manuscript has been improved. I recommend this paper for publication.

Reviewer #3 (Remarks to the Author):

The authors have addressed my comments, and I appreciate the detailed responses in their letter. I believe the revised manuscript makes a compelling case for their research, and highlights their main contributions well. I do not have further comments, and am looking forward to seeing the paper published.